# Hypoxic Conditions Promote Cartilage Repair in a Rat Knee Osteochondral Defect Model via Hypoxia-Inducible Factor-1α

**DOI:** 10.3390/ijms26136370

**Published:** 2025-07-02

**Authors:** Kei Nakamura, Atsuo Inoue, Yuji Arai, Shuji Nakagawa, Yuta Fujii, Ryota Cha, Keisuke Sugie, Kentaro Hayashi, Tsunao Kishida, Osam Mazda, Kenji Takahashi

**Affiliations:** 1Department of Orthopaedics, Graduate School of Medical Science, Kyoto Prefectural University of Medicine, Kawaramachi-Hirokoji, Kamigyo-ku, Kyoto 602-8566, Japan; keinaka2@koto.kpu-m.ac.jp (K.N.); a-inoue@koto.kpu-m.ac.jp (A.I.); y-fujii@koto.kpu-m.ac.jp (Y.F.); r-cha@koto.kpu-m.ac.jp (R.C.); k-sugie@koto.kpu-m.ac.jp (K.S.); aparagi@koto.kpu-m.ac.jp (K.H.); kenji-am@koto.kpu-m.ac.jp (K.T.); 2Department of Sports and Para-Sports Medicine, Graduate School of Medical Science, Kyoto Prefectural University of Medicine, Kawaramachi-Hirokoji, Kamigyo-ku, Kyoto 602-8566, Japan; shushi@koto.kpu-m.ac.jp; 3Department of Immunology, Graduate School of Medical Science, Kyoto Prefectural University of Medicine, Kawaramachi-Hirokoji, Kamigyo-ku, Kyoto 602-8566, Japan; tsunao@koto.kpu-m.ac.jp (T.K.); mazda@koto.kpu-m.ac.jp (O.M.)

**Keywords:** hypoxic conditions, HIF-1α, bone marrow stimulation

## Abstract

Bone marrow stimulation is a treatment for articular cartilage injuries that promotes cartilage repair by inducing the migration and accumulation of mesenchymal stem cells (MSCs), but often results in fibrocartilage with limited durability. This study aimed to investigate the effect of hypoxic conditions on cartilage repair using a rat osteochondral defect model. Osteochondral defects (1.0 mm in diameter) were created in the femoral trochlear groove, and rats were exposed to hypoxic conditions (12% O_2_) for 4 weeks postoperatively. Histological analysis was performed, and protein expression of hypoxia-inducible factor-1α (HIF-1α) and SRY-box transcription factor 9 (SOX9) in the repair tissue was evaluated after 1 week. As a result, after 1 week, protein expression of HIF-1α and SOX9 in the Hypoxia group was significantly increased compared to the Normoxia group. After 4 weeks, the Hypoxia group exhibited a hyaline cartilage-like tissue structure with a significantly lower Modified Wakitani score compared to the Normoxia group. Furthermore, after 4 weeks, the inhibition of HIF-1α suppressed cartilage repair. These findings suggest that hypoxic conditions promote SOX9 expression via HIF-1α during the early phase of MSC chondrogenic differentiation and promote the formation of hyaline cartilage-like repair tissue. In conclusion, bone marrow stimulation under hypoxic conditions may enhance the repair effect on articular cartilage injuries.

## 1. Introduction

Articular cartilage is an avascular tissue, and its capacity for self-repair is extremely limited once damaged. As cartilage lesions progress, they often lead to osteoarthritis (OA), which significantly impairs patients’ quality of life (QOL) [1]. Bone marrow stimulation, osteochondral autograft transplantation, and autologous chondrocyte implantation have been used in clinical practice to treat cartilage injuries. However, these treatments remain inadequately established, and the development of a minimally invasive technique that enables hyaline cartilage repair and maintains long-term tissue morphology and function is required [2]. Among these treatments, bone marrow stimulation has been widely used because of its simplicity and minimal invasiveness. It is generally indicated for relatively small cartilage defects of less than 2 cm^2^ and promotes cartilage repair by creating small holes in the subchondral bone at the site of injury, thereby inducing the migration and accumulation of bone marrow-derived mesenchymal stem cells (MSCs) into the site of injury [3]. However, the repair tissue formed by bone marrow stimulation is primarily fibrocartilage, which is less mechanically durable than hyaline cartilage. Therefore, this can result in difficulties in maintaining joint function in the long term [4]. Another clinical limitation is the difficulty of application to relatively large defects. One of the reasons for these issues is thought to be the insufficient chondrogenic differentiation of MSCs recruited into the repair tissue [5]. Notably, MSCs are known to be present in hypoxic stem cell niches, and recent studies have reported that hypoxic conditions promote the chondrogenic differentiation of MSCs [6]. In particular, hypoxia-inducible factor-1α (HIF-1α) has been shown to stabilize under hypoxic conditions and promote the expression of SRY-box transcription factor 9 (SOX9), a master regulator of chondrogenic differentiation, thereby contributing to chondrogenesis [7]. However, most studies on hypoxia-induced chondrogenic differentiation of MSCs have been limited to the field of cartilage tissue engineering [8]; studies on cartilage repair through the use of hypoxic conditions in vivo remain limited. Recently, hypoxic responses via HIF-1α have attracted attention in the field of musculoskeletal research, and we have previously reported the protective effects of HIF-1α on articular cartilage and its role in muscle function in vivo [9,10,11]. Given this background, we aimed to investigate the effects of hypoxic conditions on the quality of repair tissue in a rat osteochondral defect model. We hypothesized that hypoxic conditions promote the chondrogenic differentiation of MSCs, leading to the repair of hyaline cartilage-like tissue. The purpose of this study was to evaluate the effects of hypoxic conditions on cartilage repair using an animal model and to contribute to the development of novel treatment for cartilage injuries in the future.

## 2. Results

### 2.1. Effects of Hypoxic Conditions on Rat Knee Osteochondral Defect Model

Rat knee osteochondral defect models were bred under hypoxic conditions (12%), and the effects of hypoxic conditions on the repair tissue were evaluated macroscopically and histologically. Body weight in the Hypoxia group tended to decrease until postoperative day 2, but subsequently increased along the growth curve without deviation (Figure 1).

Macroscopically, both groups showed defects with surface irregularity after 1 week. After 4 weeks, the defects were repaired without depression in both groups, but the surface of the Hypoxia group was smoother than that of the Normoxia group (Figure 2a).

ICRS scores were 3.00 ± 0.00 in the Normoxia group and 3.00 ± 0.00 in the Hypoxia group after 1 week, but 6.88 ± 0.50 in the Normoxia group and 7.50 ± 0.82 in the Hypoxia group after 4 weeks, indicating a significant improvement in the Hypoxia group (*p* = 0.001) (Figure 2b).

Histologically, Safranin O staining intensity of the repair tissue after 4 weeks was higher in the Hypoxia group than in the Normoxia group. The repair tissue was mainly fibrocartilage in the Normoxia group, whereas hyaline cartilage-like tissue was observed in the Hypoxia group (Figure 3a). The Modified Wakitani score was 9.00 ± 0.00 in the Normoxia group and 9.00 ± 0.00 in the Hypoxia group after 1 week, but 4.44 ± 2.30 in the Normoxia group and 2.94 ± 1.44 in the Hypoxia group after 4 weeks, showing significantly promoted cartilage repair in the Hypoxia group (*p* = 0.04) (Figure 3b). After 4 weeks, the breakdown of the Modified Wakitani score in the Normoxia group was as follows: **cell morphology:** 2.19 ± 1.11; **matrix staining:** 1.88 ± 0.89; **thickness of cartilage:** 0.38 ± 0.62. In the Hypoxia group, these values were as follows: **cell morphology:** 1.50 ± 0.82; **matrix staining:** 1.38 ± 0.62; **thickness of cartilage:** 0.06 ± 0.25.

The repair tissues in the Hypoxia group exhibited decreased staining for Collagen Type I and increased staining for Collagen Type II compared to the Normoxia group, indicating that the repair tissue was similar to hyaline cartilage (Figure 3c). When suppression experiments were conducted using 2-Methoxyestradiol (2-ME2) as an HIF-1α-specific inhibitor, in the 2-ME2 + Hypoxia group after 4 weeks, Safranin O staining was absent, and the repair tissue was mainly fibrocartilage-like tissue (Figure 3d). The Modified Wakitani score was 8.17 ± 2.04 in the 2-ME2 + Hypoxia group, significantly higher than that in the Hypoxia group, indicating that cartilage repair was significantly suppressed by HIF-1α inhibition (*p* = 0.001) (Figure 3d).

### 2.2. The Effects of HIF-1α on the Repair Tissue

The effects of hypoxic conditions on the repair tissue were histologically evaluated using immunohistochemical staining for HIF-1α. The immunostaining area (mm^2^) of HIF-1α in the repair tissues was 0.36 ± 0.07 in the Normoxia group, 0.51 ± 0.08 in the Hypoxia group, and 0.26 ± 0.04 in the 2-ME2 + Hypoxia group after 1 week, and 0.07 ± 0.03 in the Normoxia group and 0.12 ± 0.06 in the Hypoxia group after 4 weeks. After 1 week, the protein expression of HIF-1α in the Hypoxia group was significantly increased compared to the Normoxia group (Figure 4a,b). However, after 4 weeks, the protein expression of HIF-1α in both the Normoxia and Hypoxia group was decreased (Figure 4c,d). The immunostaining area (mm^2^) of SOX9 in the repair tissues was 0.09 ± 0.03 in the Normoxia group and 0.19 ± 0.03 in the Hypoxia group, and the protein expression of SOX9 in the Hypoxia group was also significantly increased (Figure 4e,f). Furthermore, after 1 week, we observed that the protein expression of HIF-1α in the repair tissues was significantly suppressed in the 2-ME2 + Hypoxia group compared to the Hypoxia group (Figure 4a,b).

## 3. Discussion

In this study, we investigated the effect of hypoxic conditions on the repair tissue in a rat knee osteochondral defect model. The results demonstrated that at 4 weeks postoperation, the Hypoxia group exhibited repair tissue that was more similar to hyaline cartilage compared to the Normoxia group. Furthermore, at 1 week postoperation, which corresponds to the early phase after surgery, the protein expression of HIF-1α and SOX9 in the repair tissue was increased in the Hypoxia group. In addition, the repair-promoting effect of the hypoxic conditions was suppressed by an HIF-1α inhibitor (2-ME2), suggesting that hypoxic conditions may promote the chondrogenic differentiation of MSCs via HIF-1α induction, thereby contributing to the qualitative improvement in the repair tissue.

Articular cartilage is an avascular tissue with extremely limited self-repair capacity once injured. Yoshioka et al. reported that, in the knee joint of rats at 6 weeks postoperation, a V-shaped cartilage defect with a width of 0.7 mm could be repaired with hyaline cartilage-like tissue, whereas a 1.5 mm defect was predominantly repaired with fibrocartilage-like tissue [12]. Thus, while small defects may have some potential for spontaneous repair, the healing of larger defects remains challenging.

Bone marrow stimulation is a treatment that promotes cartilage repair by creating small holes in the subchondral bone at the site of injury to induce bleeding from the bone marrow, leading to the formation of a marrow clot containing MSCs, which serves as the basis for inducing cartilage repair. However, the repair tissue is generally fibrocartilage, which has inferior mechanical properties compared to hyaline cartilage, and may eventually lead to secondary osteoarthritis (OA) in the long term [3]. One of the possible reasons for this qualitative problem is the insufficient chondrogenic differentiation of MSCs during the repair process [4,5]. Recently, the application of hypoxia-responsive mechanisms in cartilage repair has garnered significant attention. Articular cartilage is physiologically under hypoxic conditions of 1–8% [13], and many studies have reported the effects of hypoxic conditions on chondrocytes and the chondrogenic differentiation of MSCs. Nevo et al. were the first to demonstrate that hypoxic conditions promote the differentiation of embryonic chick chondrocytes [14]. Henrionnet et al. reported that hypoxic conditions represent a critical environmental factor that promotes chondrogenic differentiation while inhibiting the osteogenic differentiation of MSCs [15]. Semenza et al. identified hypoxia-inducible factor-1α (HIF-1α) as a key regulator of hypoxic responses, demonstrating that HIF-1α is stabilized under hypoxic conditions by escaping degradation via prolyl hydroxylases (PHDs) and functions as a transcription factor [16,17,18]. Amarilio et al. demonstrated that HIF-1α directly binds to the promoter region of SOX9 and promotes chondrogenic differentiation [7], and many reports have shown that hypoxia enhances the chondrogenic differentiation potential of MSCs in vitro. For example, Adesida et al. found that culturing human bone marrow-derived MSCs under hypoxic conditions upregulated HIF-1α expression and increased the expression of chondrogenic markers such as SOX9, ACAN, and COL2A1 [19]. Duval et al. demonstrated that hypoxia-preconditioned MSCs exhibit enhanced extracellular matrix production [20]. Li et al. revealed that hypoxic conditions promote chondrogenic differentiation while suppressing the hypertrophic differentiation of MSCs [21]. Shimomura et al. showed similar effects in induced pluripotent stem (iPS) cells [22]. Although these findings have been mainly based on in vitro studies in the field of cartilage tissue engineering, this study provides important in vivo evidence that hypoxic conditions promote chondrogenic differentiation within the repair tissue of osteochondral defects.

In this study, we adopted hypoxic conditions with an oxygen concentration of 12%, which has previously been confirmed to be safe and effective in the musculoskeletal field [9,10]. In this environment, 1.0 mm diameter osteochondral defect models were created, and histological evaluation was performed at 4 weeks postoperation. As a result, the Hypoxia group exhibited enhanced Safranin O staining, upregulated protein expression of Collagen Type II, and decreased protein expression of Collagen Type I, indicating the formation of a tissue structure closer to that of hyaline cartilage. Furthermore, at 1 week postoperation, the protein expression of HIF-1α and SOX9 was increased in the Hypoxia group. Inhibition of HIF-1α with 2-ME2 led to the suppression of the increased expression of these proteins, resulting in the formation of fibrocartilage-like tissue, suggesting that HIF-1α plays a pivotal role in promoting cartilage repair under hypoxic conditions.

HIF-1α forms a complex with ARNT (HIF-1β) in the cytoplasm and subsequently enters the nucleus, where it directly promotes the transcription of SOX9 by binding to the hypoxia-responsive element located in the promoter region of the SOX9 gene. In addition, it is also known that HIF-1α regulates SOX9 expression by activating the TGF-β/Smad, BMP/Smad, and Hedgehog pathways, while inhibiting the Notch and Wnt/β-catenin pathways [23]. These pathways interact with each other, and SOX9, in conjunction with SOX5 and SOX6, forms the SOX trio to promote the transcription of the ACAN and COL2A1 genes. Yan et al. demonstrated that the TGF-β/Smad pathway is activated via HIF-1α under hypoxic conditions [24], while Tan et al. demonstrated the presence of crosstalk between HIF-1α and the Wnt/β-catenin pathway [25]. Based on these findings, in this study, it is highly likely that HIF-1α induced in MSCs within the repair tissue under hypoxic conditions directly or indirectly promoted SOX9 transcriptional activity, thereby contributing to the formation of hyaline cartilage-like repair tissue.

### Limitations

This study has several limitations. First, the experiment was conducted under a single hypoxic condition, and the effects of different oxygen concentrations were not evaluated. Second, the evaluation period was limited to the short term, with assessments performed only up to 4 weeks after surgery. Third, the HIF-1α inhibitor (2-ME2) used in this study does not act specifically on the repair tissue, and the potential effects of the inhibitor itself on the repair tissue remain unclear. Finally, the downstream signaling pathways of HIF-1α in MSCs within the repair tissue have not been fully elucidated.

## 4. Materials and Methods

### 4.1. Animals

A total of fifty-four 6-week-old male Wistar rats (Shimizu Laboratory Suppliers, Kyoto, Japan) were used in this study. The study was approved by the Institutional Animal Care and Use Committee of our institution (code: M2023-286) and conducted in accordance with its ethical guidelines.

### 4.2. Creation of Rat Knee Osteochondral Defect Model

Anesthesia induction was performed by administering 0.375 mg/kg medetomidine, 2.0 mg/kg midazolam, and 2.5 mg/kg butorphanol intraperitoneally to the rats. An anterior straight longitudinal incision was made on the left knee joint, and the knee was exposed using a medial parapatellar approach. The patella was dislocated laterally to expose the femoral trochlear groove. According to the method reported by Yoshioka et al., which is known to create an osteochondral defect that does not allow for spontaneous repair with hyaline cartilage, an osteochondral defect (1.0 mm in diameter and 2.0 mm in depth) reaching the subchondral bone was created in the femoral trochlear groove using a Kirschner wire (diameter: 1.0 mm; ISO Medical Advance, Osaka, Japan), and bleeding from the bone marrow was confirmed [12]. After reduction of the patella, the joint capsule and skin were sutured using 5-0 nylon sutures to complete the surgery.

### 4.3. Breeding in a Hypoxic Chamber

Hypoxic conditions were established using a chamber with an adjustable oxygen concentration (Natsume Seisakusho Co., Ltd., Wakenyaku Co., Ltd., Kyoto, Japan), and the equipment enabled breeding under hypoxic conditions. Nitrogen supplied from a gas generator was mixed with ambient air at a desired ratio using an air blender to reduce the oxygen concentration. This hypoxic air was circulated through the breeding cage and workplace to maintain hypoxic conditions in the chamber. The oxygen concentration in any given area of the chamber could be measured using a gas analyzer, and the oxygen concentration in the chamber could be adjusted at will within the range of 4–20% by adjusting the air blender (Figure 5). Based on previous reports of animal models, the oxygen concentration was set at 12% in the hypoxic condition [9,10].

### 4.4. Effects of Hypoxic Conditions on Rat Knee Osteochondral Defect Model

Postoperatively, rats were divided into the Normoxia group (*n* = 20, 21% oxygen) and the Hypoxia group (*n* = 20, 12% oxygen). At 1 and 4 weeks after surgery (1 week: *n* = 4 per group; 4 weeks: *n* = 16 per group), rats were anesthetized with pentobarbital, and their left femur was surgically removed (Figure 6). Both groups were bred under a 12 h light/dark cycle with free access to food and water. The body weight of both groups was measured over time until 4 weeks postoperation (*n* = 16).

### 4.5. HIF-1α Inhibitor Treatment

To evaluate the effects of HIF-1α under hypoxic conditions, inhibition of HIF-1α was performed. 2-ME2 (Selleck, Tokyo, Japan) was used as an HIF-1α-specific inhibitor. After creating an osteochondral defect in the left femoral trochlear groove of rats (*n* = 10), rats were bred under 12% hypoxic conditions, and 2-ME2 was intraarticularly administered three times per week starting from the day after surgery, according to previous reports, at an appropriate concentration (100 μM) [26]. The animals were defined as the 2-ME2 + Hypoxia group. Their left femur was surgically removed at 1 week (*n* = 4) and 4 weeks (*n* = 6) after surgery.

### 4.6. Macroscopic Evaluation

Macroscopic evaluation of the osteochondral defects in the femoral trochlear groove was performed at 1 week (*n* = 4 per group) and 4 weeks (*n* = 16 per group) postoperation in the Normoxia and Hypoxia groups. The repair tissue was evaluated using the International Cartilage Repair Society (ICRS) scoring system [27]. In this study, to focus on the quality of the repair tissue, the item “Integration to border zone” was excluded from the evaluation, and scoring was performed on an 8-point scale.

### 4.7. Histological Evaluation

After macroscopic evaluation, samples were fixed in 4% paraformaldehyde (Wako, Osaka, Japan), decalcified in 20% EDTA, and embedded in paraffin. Sagittal sections (6 μm thick) were prepared from the created defect site and stained with Safranin O. Histological evaluation of the repair tissue in the femoral trochlear groove was performed under a microscope at 1 week (*n* = 4 per group) and 4 weeks (*n* = 16 per group) postoperation in the Normoxia and Hypoxia groups. The repair tissue was evaluated using the Modified Wakitani score [28]. In addition, the effect of HIF-1α was evaluated by comparing the Hypoxia group (*n* = 16) with the 2-ME2 + Hypoxia group (*n* = 6) at 4 weeks postoperation. In this study, to focus on the quality of the repair tissue, the item “Integration of implant with adjacent host cartilage” was excluded from the evaluation, and scoring was performed on a 9-point scale. The components of the score were as follows: cell morphology (4 points), matrix staining (3 points), and thickness of cartilage (2 points).

### 4.8. Immunohistochemical Analysis

Immunohistochemical analysis was performed to evaluate the expression of HIF-1α (1 week: *n* = 4 per group; 4 weeks: Normoxia and Hypoxia groups, *n* = 2), SOX9 (1 week: Normoxia and Hypoxia groups, *n* = 4), Collagen Type I, and Collagen Type II (4 weeks: Normoxia and Hypoxia groups, *n* = 1). Paraffin-embedded sections were deparaffinized with xylene, rehydrated through graded ethanol, and washed with running water and PBS. Endogenous peroxidase activity was blocked via incubation in 3% H_2_O_2_ methanol solution for 15 min. The sections were incubated overnight at 4 °C with rabbit polyclonal anti-HIF-1α antibody (ab114977; Abcam, Cambridge, UK; 1:100 dilution), mouse monoclonal anti-SOX9 antibody (ab76997; Abcam, Cambridge, UK; 1:1200 dilution), rabbit polyclonal anti-Collagen Type I antibody (ab34710; Abcam, Cambridge, UK; 1:40 dilution), and mouse monoclonal anti-Collagen Type II antibody (F-57; Kyowa Pharma Chemical Co., Toyama, Japan; 1:100 dilution). After washing with PBS, the sections were incubated with Histofine^®^ Simple Stain Rat MAX-PO (Nichirei Biosciences Inc., Tokyo, Japan) at 25 °C for 30 min. Immunostaining was detected by using DAB staining, and Mayer’s hematoxylin was used for counterstaining. The stained sections were observed under a light microscope. Quantitative analysis of the expression of HIF-1α and SOX9 was performed using ImageJ (version 1.54g). For each sample, the region of interest was defined as the entire osteochondral defect site created in the femoral trochlear groove. Only the repair tissue within the defect area was analyzed. First, the images were converted to 8-bit grayscale, and color deconvolution was applied to separate DAB staining from hematoxylin. The DAB channel was then selected, and thresholding was applied to identify positively stained areas. The area of positive staining was subsequently measured.

### 4.9. Statistical Analysis

All data were expressed as the mean ± standard error of the mean (SEM). Statistical analysis was performed using EZR (Saitama Medical Center, Jichi Medical University, Saitama, Japan), a graphical user interface for R (The R Foundation for Statistical Computing, Vienna, Austria). The Shapiro–Wilk test was used to assess the normality of data distribution. For datasets that did not follow a normal distribution (*p* < 0.05), the nonparametric Mann–Whitney U test was used. For normally distributed data, the unpaired Student’s *t*-test was applied. The statistical significance level was set at *p* < 0.05.

## 5. Conclusions

This study demonstrated that applying a 12% hypoxic condition to a rat osteochondral defect model may increase HIF-1α expression in MSCs within the repair tissue, promote their chondrogenic differentiation, thereby contributing to the formation of hyaline cartilage-like tissue. Bone marrow stimulation under hypoxic conditions may enhance the repair effect on articular cartilage injuries.

## Figures and Tables

**Figure 1 ijms-26-06370-f001:**
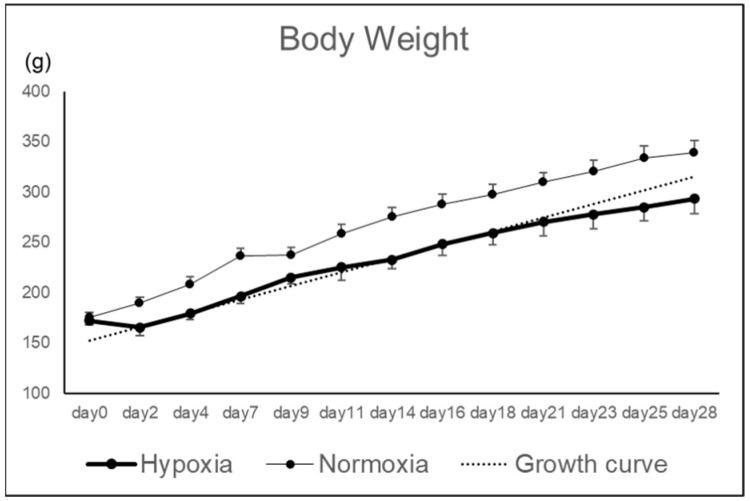
Body weight.

**Figure 2 ijms-26-06370-f002:**
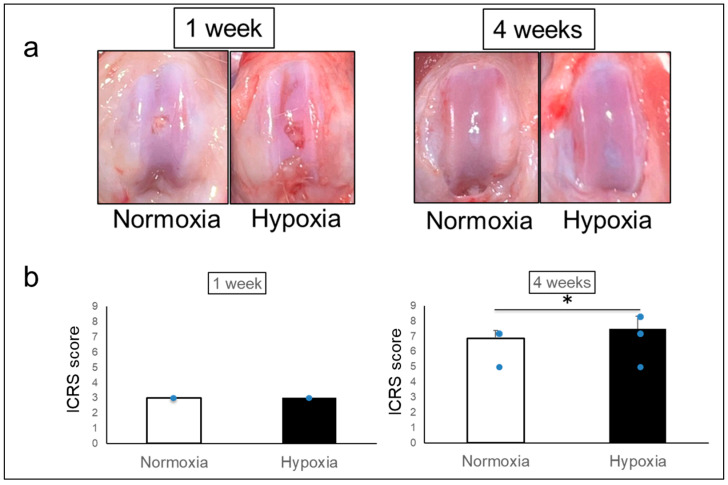
Macroscopic findings. (**a**) Representative photographs. (**b**) ICRS score (mean ± standard deviation). * *p* < 0.05 (1 week: Normoxia group, *n* = 4; Hypoxia group *n* = 4) (4 weeks: Normoxia group, *n* = 16; Hypoxia group, *n* = 16).

**Figure 3 ijms-26-06370-f003:**
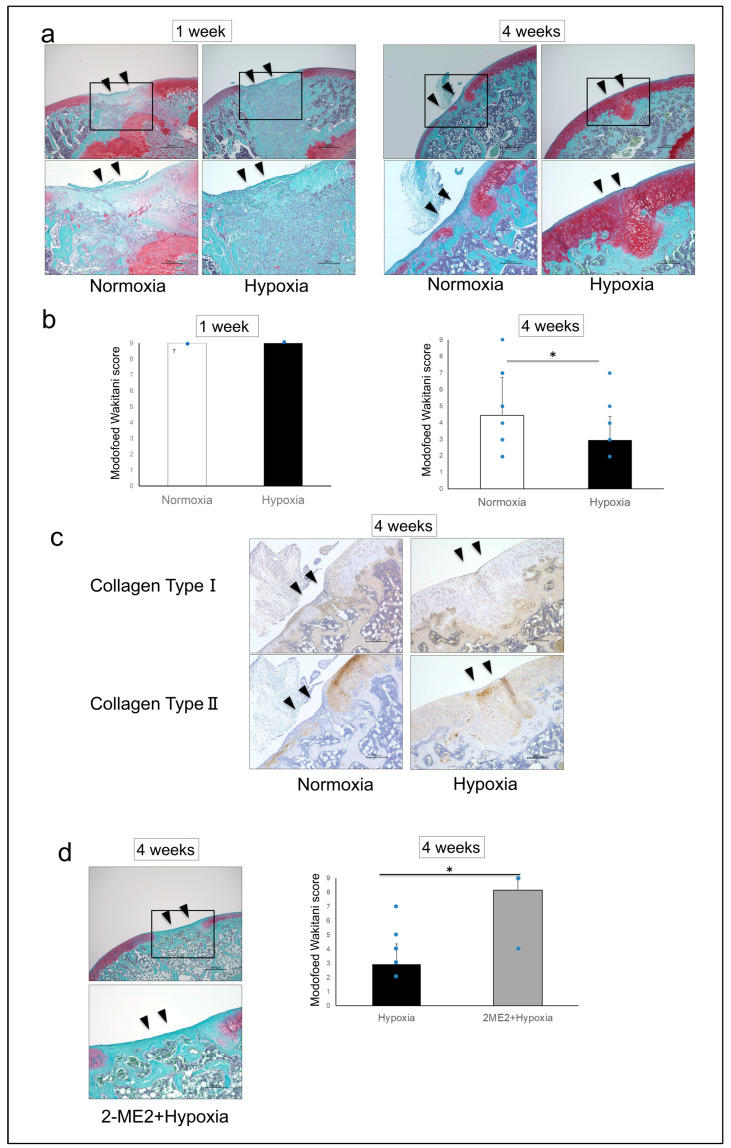
(**a**) Representative micrographs of Safranin O-stained sagittal sections and fragments (scale bar: upper 500 μm, lower 200 μm), (**b**) the Modified Wakitani score (mean ± standard deviation) (1 week: Normoxia group, *n* = 4; Hypoxia group *n* = 4. 4 weeks: Normoxia group, *n* = 16; Hypoxia group, *n* = 16), (**c**) micrographs of Collagen Type I and Collagen Type II immunostaining (scale bar: 200 μm), (**d**) representative micrographs of Safranin O-stained sagittal sections of the 2-ME2 + Hypoxia group (scale bar: upper 500 μm, lower 200 μm), the Modified Wakitani score (mean ± standard deviation) compared to the Hypoxia group (* *p* < 0.05) (Hypoxia group, *n* = 16; 2-ME2 + Hypoxia group, *n* = 6). ▼: the area of the osteochondral defect. Black boxes in the 40× images indicate the regions shown at higher magnification (100×).

**Figure 4 ijms-26-06370-f004:**
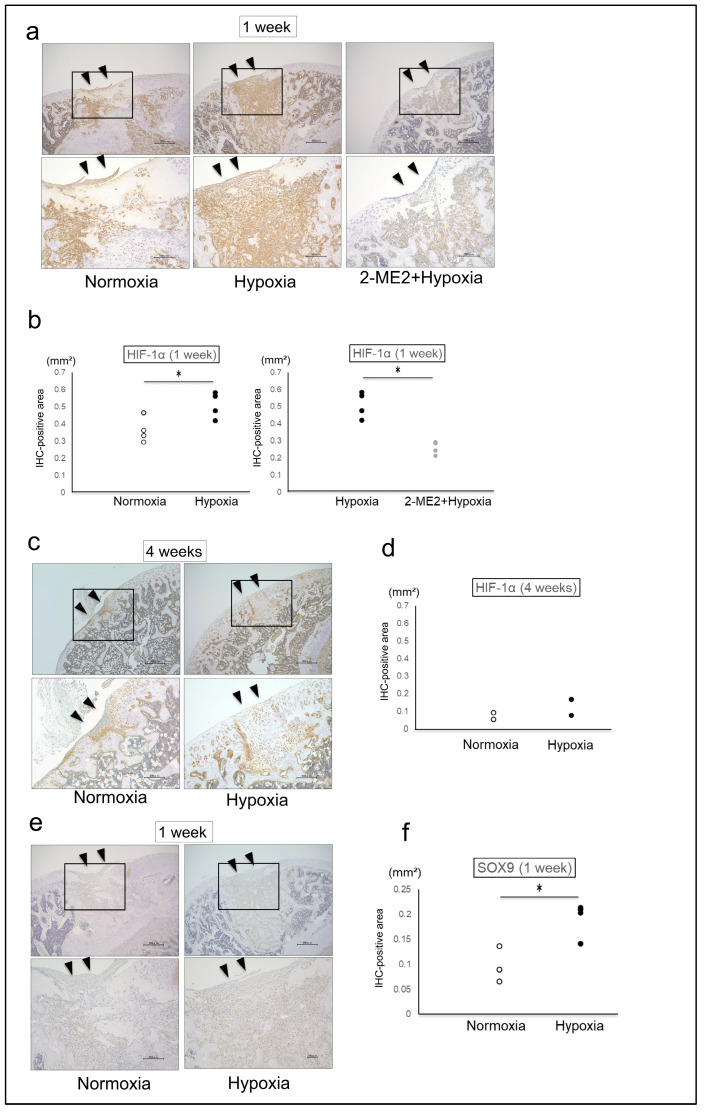
(**a**) HIF-1α immunostaining at 1 week postoperation, with a representative micrograph scale bar as follows: upper 500 μm, lower 200 μm. (**b**) HIF-1α immunostaining at 1 week postoperation (Normoxia group, *n* = 4; Hypoxia group, *n* = 4; 2-ME2 + Hypoxia group, *n* = 4). (**c**) HIF-1α immunostaining at 4 weeks postoperation, with representative micrographs (scale bar: upper 500 μm, lower 200 μm). (**d**) HIF-1α immunostaining at 4 weeks postoperation (Normoxia group, *n* = 2; Hypoxia group, *n* = 2). (**e**) SOX9 immunostaining at 1 week postoperation, with representative micrographs (scale bar: upper 500 μm, lower 200 μm). (**f**) SOX9 immunostaining at 1 week postoperation (Normoxia group, *n* = 4; Hypoxia group, *n* = 4). The IHC-positive area was quantified using ImageJ software (version 1.54g). The region of interest was defined as the entire osteochondral defect (▼) created in the femoral trochlear groove, and only the repair tissue within this area was analyzed. ▼: the area of the osteochondral defect. Black boxes in the 40× images indicate the regions shown at higher magnification (100×). An asterisk (*) indicates *p* < 0.05.

**Figure 5 ijms-26-06370-f005:**
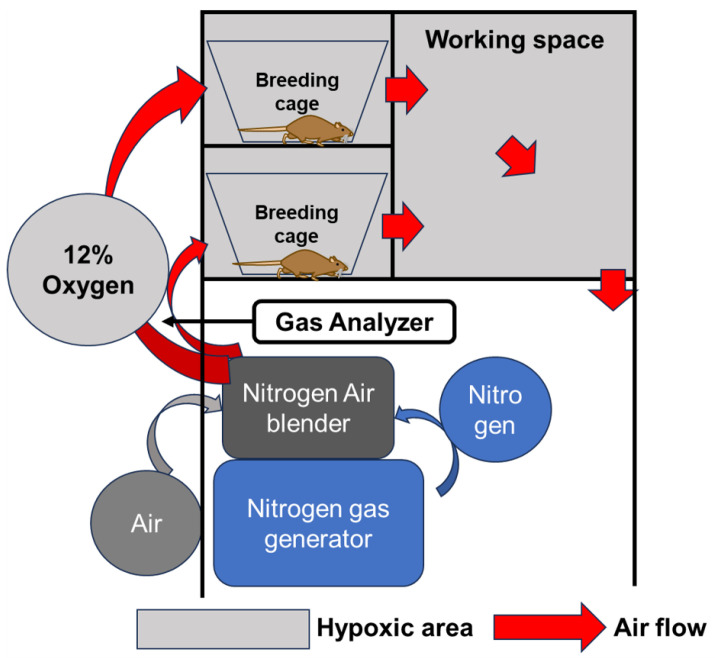
Nitrogen generated by the N_2_^+^ gas generator is mixed with outside air using N_2_^+^ air blender to induce hypoxic conditions. The concentration in the chamber can be adjusted as desired by circulating low oxygen levels through the chamber.

**Figure 6 ijms-26-06370-f006:**
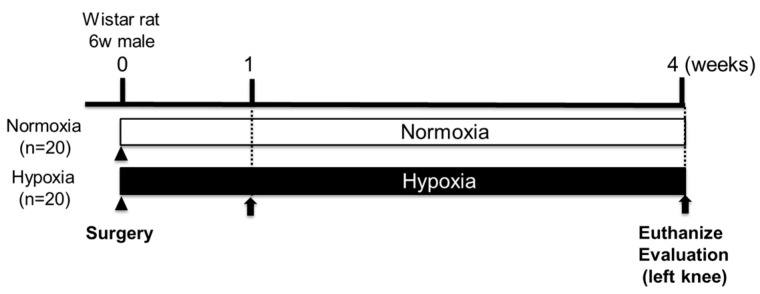
The protocol for the evaluation of cartilage repair under hypoxic conditions. The arrow (➡) represents the timeline from animal euthanasia to subsequent evaluation.

## Data Availability

The datasets used and/or analyzed during the current study are available from the corresponding author on reasonable request.

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
