# Peer review of "Hypoxic Conditions Promote Cartilage Repair in a Rat Knee Osteochondral Defect Model via Hypoxia-Inducible Factor-1α"

_ijms, 2025, doi:10.3390/ijms26136370_

Round 1
Reviewer 1 Report
Comments and Suggestions for Authors
The authors evaluated the effect of hypoxia on cartilage repair in a rat osteochondral defect model, demonstrating that hypoxic conditions promote early chondrogenic differentiation and the formation of hyaline-like cartilage. However, several points still to be addressed.
- The resolution of the figures should be improved to enhance the visualization of key features. Additionally, scatter plots with bars are recommended to display individual data points and provide a clearer representation of data distribution.
- Please specify the number of animals or samples analyzed for each group in the figure legend of all the figures to enhance clarity and reproducibility.
- For the Wakitani Scoring used in 2.1, it would be helpful to provide a more detailed breakdown of the Wakitani cartilage repair scoring across different subcategories (e.g., cell morphology, matrix staining, surface regularity, etc.).
- It is recommended to increase the number of samples used for IHC staining to ensure statistical reliability, as using only two samples per group is insufficient to draw robust or conclusive results. Additionally, in the Methods section, please provide a more detailed description of how ImageJ was used for quantification of IHC staining (e.g., color deconvolution, thresholding, area or intensity measurement)
- In line 94, "The repair tissues in the Hypoxia group exhibited decreased staining for Collagen Type I and increased staining for Collagen Type II," please specify compared to which group these changes were observed (e.g., Normoxia or control group).
The English expression needs improvement to make it easier for readers to understand.
Reviewer 2 Report
Comments and Suggestions for Authors
- This study demonstrates that hypoxic conditions may increase HIF-1α expression in MSCs within the repair tissue and promote their chondrogenic differentiation in a rat knee osteochondral defect model. The study is well designed and structured. However, there are a few points that need revision.
- The background section in the Abstract is relatively lengthy. It is recommended to shorten this part and expand the description of the results to better reflect the key findings.
- In lines 75–76, did the body weight in the Hypoxia group tend to decrease until postoperative day 2 rather than day 4? Please confirm.
- The control group size (n=2) at the 1-week timepoint appears too small to draw statistically meaningful conclusions. It is necessary to address the limitations in the interpretation of the 1-week results in the Discussion section.
- It is recommended to add a paragraph addressing the limitations of the study in the Discussion section.
Round 2
Reviewer 1 Report
Comments and Suggestions for Authors
There are two points that require revision:
-
All bar graphs should include clear labels on the Y-axis indicating the specific parameter being analyzed.
-
The method used to quantify the IHC-positive area needs to be clarified further. Specifically, in Figure 4a, please indicate which exact region was analyzed.
Reviewer 2 Report
Comments and Suggestions for Authors
The manuscript has been adequately revised and is now suitable for publication. I have no further comments.
